# Identification of the Transcription Factor ATF3 as a Direct and Indirect Regulator of the LDLR

**DOI:** 10.3390/metabo12090840

**Published:** 2022-09-06

**Authors:** Sabine Bauer, Jana Eigenmann, Yuqi Zhao, Julia Fleig, Johann S. Hawe, Calvin Pan, Dario Bongiovanni, Simon Wengert, Angela Ma, Aldons J. Lusis, Jason C. Kovacic, Johan L. M. Björkegren, Lars Maegdefessel, Heribert Schunkert, Moritz von Scheidt

**Affiliations:** 1Department of Cardiology, German Heart Centre Munich, Technical University Munich, 80636 Munich, Germany; 2DZHK (German Centre for Cardiovascular Research), Partner Site Munich Heart Alliance, 80802 Munich, Germany; 3Department of Integrative Biology and Physiology, Institute for Quantitative and Computational Biosciences, David Geffen School of Medicine, University of California, Los Angeles, CA 90095, USA; 4Departments of Medicine, Human Genetics, Microbiology, Immunology and Molecular Genetics, David Geffen School of Medicine, University of California, Los Angeles, CA 90095, USA; 5Division of Cardiology, Cardiocentro Ticino Institute, Ente Ospedaliero Cantonale, 6900 Lugano, Switzerland; 6Helmholtz Pioneer Campus, Helmholtz Center Munich, 85764 Neuherberg, Germany; 7Institute of Computational Biology, Helmholtz Center Munich, 85764 Oberschleißheim, Germany; 8Department of Genetics and Genomic Sciences, Institute of Genomics and Multiscale Biology, Icahn School of Medicine at Mount Sinai, New York, NY 10029, USA; 9Victor Chang Cardiac Research Institute, Darlinghurst, Sydney, NSW 2010, Australia; 10St. Vincent’s Clinical School, University of New South Wales, Darlinghurst, Sydney, NSW 2010, Australia; 11Icahn School of Medicine at Mount Sinai, Cardiovascular Research Institute, New York, NY 10029, USA; 12Clinical Gene Networks AB, 114 44 Stockholm, Sweden; 13Integrated Cardio Metabolic Centre, Karolinska Institutet, Novum, Huddinge, 171 77 Stockholm, Sweden; 14Department of Vascular and Endovascular Surgery, Klinikum Rechts der Isar, Technical University Munich, 81675 Munich, Germany; 15Department of Molecular Medicine and Surgery, Karolinska Institutet, 171 76 Stockholm, Sweden

**Keywords:** ATF3, atherosclerosis, cardiovascular disease, coronary artery disease, gene expression, inflammation, LDLR, lipid metabolism, liver metabolism, LPS, MAFF, transcription factor

## Abstract

Coronary artery disease (CAD) is a complex, multifactorial disease caused, in particular, by inflammation and cholesterol metabolism. At the molecular level, the role of tissue-specific signaling pathways leading to CAD is still largely unexplored. This study relied on two main resources: (1) genes with impact on atherosclerosis/CAD, and (2) liver-specific transcriptome analyses from human and mouse studies. The transcription factor activating transcription factor 3 (ATF3) was identified as a key regulator of a liver network relevant to atherosclerosis and linked to inflammation and cholesterol metabolism. ATF3 was predicted to be a direct and indirect (via MAF BZIP Transcription Factor F (MAFF)) regulator of low-density lipoprotein receptor (LDLR). Chromatin immunoprecipitation DNA sequencing (ChIP-seq) data from human liver cells revealed an ATF3 binding motif in the promoter regions of MAFF and LDLR. siRNA knockdown of ATF3 in human Hep3B liver cells significantly upregulated LDLR expression (*p* < 0.01). Inflammation induced by lipopolysaccharide (LPS) stimulation resulted in significant upregulation of ATF3 (*p* < 0.01) and subsequent downregulation of LDLR (*p* < 0.001). Liver-specific expression data from human CAD patients undergoing coronary artery bypass grafting (CABG) surgery (STARNET) and mouse models (HMDP) confirmed the regulatory role of ATF3 in the homeostasis of cholesterol metabolism. This study suggests that ATF3 might be a promising treatment candidate for lowering LDL cholesterol and reducing cardiovascular risk.

## 1. Introduction

Coronary artery disease (CAD) remains one of the leading causes of death and morbidity worldwide [1,2] and is caused by atherosclerosis of the coronary arteries. Atherosclerosis is a chronic inflammatory disease affecting arteries and caused by inflammation and lipid accumulation, as well as smooth muscle cell differentiation resulting in the formation of necrotic cores [3,4,5]. The accumulation of low-density lipoprotein (LDL) cholesterol, triglycerides and fatty acids due to obesity, western lifestyle and insulin resistance is one of the major risk factors for cardiovascular disease [6,7,8]. This is significantly driven by perturbation of cholesterol metabolism and inflammation [9,10,11]. Genome-wide association studies (GWAS) in patients with CAD or myocardial infarction (MI), as well as genetic mouse models of atherosclerosis, revealed a strong correspondence of signaling pathways and networks relevant to atherosclerosis [12,13,14,15,16]. For example, Mendelian randomization studies confirmed the promotion of atheroprogression and CAD through an increase in plasma cholesterol and inflammation anchored primarily in liver metabolism [8,17,18]. The underlying mechanisms are still inadequately understood. 

The LDL receptor (LDLR) is a central gene involved in the regulation of the LDL cholesterol homeostasis and progression of atherosclerosis. It is the main receptor for hepatic uptake of plasma LDL and mutations in the LDLR have been identified in familial hypercholesterolemia and premature atherosclerosis [17,18]. The expression of the LDLR can be regulated at both the transcriptional and translational levels [19,20,21]. A major objective of atherosclerosis research is to identify regulators of the LDLR that are suitable for therapeutic applications. In previous work, this research group was already able to demonstrate that the transcription factor MAF BZIP Transcription Factor F (MAFF) regulates the expression of the LDLR and other genes known to influence atherosclerosis and CAD in context-specific manner. MAFF was identified as a missing link between inflammation and lipid metabolism [22]. 

Activating transcription factor (ATF) 3 is a stress-induced transcription factor involved in the modulation of metabolic, inflammatory and oncogenic processes [23,24,25,26,27]. With a focus on the development of atherosclerosis, ATF3 plays a crucial role in a multitude of pathophysiological changes, such as extracellular matrix dysfunction, smooth muscle cell proliferation and migration, and foam cell formation [28]. ATF3 belongs to the ATF/cAMP response element-binding family of leucine zipper transcription factors and binds to the consensus sequence TGACGTCA in the promoter region of genes [29,30] regulating transcription through homodimerization and heterodimerization with other bZiP proteins by repressing and inducing gene expression [31]. ATF3 expression has been shown to be induced by inflammatory chemo- and cytokines, oxLDL and lipopolysaccharides, and is involved in the regulation of apoptosis and cell death shown in vascular endothelial cells and macrophages [32,33]. Upregulation of ATF3 levels has been reported in human atherosclerotic arteries compared to a healthy phenotype [32]. ATF3 has been linked to atherosclerosis in different tissues and discussed as a potential biomarker and protective factor against atherosclerosis [27,34,35]. Recently, an atheroprotective mechanism of ATF3 via regulation of high-density lipoprotein (HDL) and bile acid metabolism was identified [25]. However, the regulatory capacities of ATF3 on atherosclerosis relevant pathways remain incomplete.

For this, the main focus of this work, as well as previous studies by this group, has been on the analysis of liver-specific regulatory networks that are predicted to impact the development and progression of atherosclerosis [22].

## 2. Results

### 2.1. Identification of ATF3 As a Central Key Driver Gene of a Liver-Specific Regulatory Network

The workflow of this study is summarized in Figure 1. 

Liver-specific gene regulatory networks were modeled separately based on 244 human CAD GWAS candidate genes and 827 genes that have already been validated to have a significant effect on atherosclerosis in genetically engineered mouse models [34,35,36,37,38,39,40,41,42]. The list of candidate genes and detailed methods have been published as a supplement file before [22]. Bayesian network models based on publicly accessible gene expression data from liver-specific human and mouse studies were used to separately model the regulatory capacity between genes in the network for both species. Human CAD GWAS candidate genes and mouse atherosclerosis genes were subsequently mapped to the liver-specific Bayesian network model to identify subnetworks of interest, highly enriched with atherosclerosis and CAD relevant genes. Based on this bioinformatics approach the following key driver genes were identified in this dense liver-specific gene regulatory network (ABCG5, ABCG8, ALDH2, ATF3, CCL7, COL6A3, CXCL10, EGR2, FCER1G, IGSF6, IL15RA, IL1B, ITGAL, LOXL2, LTB, MAFF, NGRN, SERPINE1, TLR2 and ZNF467). After combining both models into a unified liver-specific Bayesian network, focus was on the subnetwork of the central key driver gene and transcription factor ATF3 (FDR 3.89 × 10^−6^; Fold enrichment 14.05). The identified network of ATF3 consists of 50 genes. Among those 17 genes have already been shown to have a significant effect on atherosclerosis either as human CAD GWAS candidate gene (LDLR, PLAUR, RGS1, TRIB1 and TSC22D2) or in engineered mouse models (Ccl7, Edn1, Erg1, Erg2, Gdf15, Il1b, Ldlr, Mmp12, Nr4a1, Nr4a2, Plaur, Rgs1, Serpine1, Sgk1, Trib1, Tsc22d2, Vcam1) [18,22,33,43,44,45,46,47,48,49,50,51,52,53,54,55]. To further elucidate the regulatory capacities and downstream effects of ATF3 on atherosclerosis and CAD relevant genes (e.g., MAFF and LDLR), hierarchical information was implemented in the following step.

### 2.2. Identification of ATF3 As Direct and Indirect Regulator of LDLR

Based on several gene expression datasets from mouse and human studies to modulate Bayesian network models, a dense liver subnetwork was identified and predicted to be orchestrated by ATF3 (Figure 2). The regulatory direction between genes based on expression data is subject to different environmental influences in complex networks. To account for different physiological aspects of the included studies, a consensus directionality was implemented based on the numeric majority of supporting studies. This implies that all predicted regulatory directionalities can potentially be bidirectional. 

Most importantly, this bioinformatics approach predicted ATF3 to be upstream of MAFF and the LDLR with the potential ability to regulate LDLR expression in a direct and via MAFF in an indirect manner. EDN1, EGR2, GDF15, IL1B, LDLR, MAFF, PLAUR, RGS1, SERPINE1, SGK1 and TSC22D2 were predicted to be downstream whereas CCL7, EGR1, MMP12, NR4A1, NR4A2, TRIB1 and VCAM1 were predicted to be upstream of ATF3. Regarding the regulatory capacities of MAFF—EPHA2, GDF15, LDLR, MCL1, PHLDA1, TNFAIP3 and TNFRSF12A were predicted to be downstream of MAFF and ATF1, SERPINE1, TRIB1 and ZFP36 were predicted to be upstream of MAFF. To confirm the bioinformatics prediction of regulatory capacities, screening of potential ATF3 binding sites on MAFF and LDLR based on human liver cells was attempted.

### 2.3. ATF3 Has the Potential to Bind to MAFF and LDLR Promoter and Intronic Regions

The identification of ATF3 binding sites in promoter and intron regions of LDLR and MAFF was based on chromatin immunoprecipitation sequencing (ChIP-seq) data from human HepG2 liver cells. Two replicates were processed using the established Encyclopedia of DNA Elements (ENCODE) Transcription Factor ChIP-seq processing pipeline. Significant ChIP-seq signals with p-values below 10 × 10^−5^ were taken into consideration and further inspected. Overall, 15,412 conserved peaks were identified between the replicates with an irreproducible discovery rate (IDR) threshold of <0.05. Figure 3 provides most reliable ATF3 binding sites in the promoter and introns regions of MAFF (A) and LDLR (B) as well as the best matching binding site motif (C).

### 2.4. In Vitro Experiments Clarify the Regulatory Role of ATF3 on MAFF and LDLR Expression 

To experimentally confirm the role of ATF3 affecting the predicted regulatory liver network, human Hep3B cells were studied in vitro. Hep3B cells are a common hepatocyte cell line that cover most relevant metabolic aspects of the human liver well. Small interfering RNA (siRNA) knockdown of ATF3 revealed consistent and significant downregulation of ATF3 after 48 h (80%, *p* < 0.001). Non-coding scramble siRNA knockdown was used as a control treatment. Regarding perturbation of predicted downstream targets, siRNA knockdown of ATF3 in human Hep3B cells showed highly significant upregulation of LDLR expression (*p* < 0.001), as well as significant upregulation of MAFF expression (*p* < 0.01) (Figure 4). In addition, other genes downstream of MAFF (e.g., SERPINE1) showed significant upregulation by downregulation of ATF3 (data not shown).

### 2.5. Inflammatory Effects on ATF3, LDLR and MAFF Expression in Liver Tissue

To assess the effects of inflammation on expression in the ATF3-regulated subnetwork, stimulation with lipopolysaccharide (LPS) was performed in human Hep3B cells. Liver cells were treated with 100 ng/mL LPS for 24 h. ATF3 expression was significantly upregulated after LPS stimulation compared to controls without LPS stimulation (*p* < 0.01) while LDLR expression was significantly downregulated (*p* < 0.0001) and MAFF expression was significantly increased after LPS stimulation compared to untreated controls (*p* < 0.0001) (Figure 5).

### 2.6. In Vivo Data of Human and Mouse Confirm the Importance of ATF3 for Cholesterol Metabolism 

In vivo validation in humans was based on transcriptomic data from the STARNET (Stockholm-Tartu Atherosclerosis Reverse Network Engineering Task) study. STARNET comprises genetic and transcriptomic data from more than 600 CAD patients undergoing coronary artery bypass grafting (CABG) surgery and represents the most important repository of its kind. Transcriptomic data are available for blood, internal mammary artery, atherosclerotic aortic root, subcutaneous fat, visceral abdominal fat, skeletal muscle and liver. These data are used to study, amongst others, co-expression networks, gene regulatory networks and supernetworks capturing multiple scales of disease progression [56]

The focus of this study was to elucidate the role of ATF3 in human liver metabolism. Based on the bioinformatics modeling and in vitro validation ATF3 expression was correlated to its predicted downstream targets. ATF3 showed significant positive correlation with LDLR (Pearson’s R 0.65, *p* = 4.33 × 10^−67^) and MAFF (Pearson’s R 0.67, *p* = 1.55 × 10^−72^) expression in human liver (Figure 6A,B).

Moreover, ATF3 was identified as a key regulator of ’liver module 15’ (Appendix A)—orchestrating an important supernetwork associated with obesity, diabetes, cholesterol metabolism, inflammation and atherosclerosis. Several CAD relevant phenotypes showed significant associations with this ATF3 regulated liver module. Most importantly, plasma CRP (a measure for inflammation) (*p* < 1.0 × 10^−100^), fasting plasma LDL cholesterol (*p* < 1.0 × 10^−100^), total cholesterol (*p* < 1.0 × 10^−100^) and atherosclerotic lesions (*p* < 1.0 × 10^−59^) showed significant correlation (Figure 6C). 

Pathway analysis based on Gene Ontology (GO) terms revealed most relevant biological pathways, involved cell compartments and molecular function for the liver supernetwork. Significant upregulation of inflammatory pathways (response to cytokines, cellular response to cytokine stimulus and response to biotic stimulus; *p* < 1.0 × 10^−75^ each) was identified (Figure 6D). 

The Hybrid Mouse Diversity Panel (HMDP) comprises 105 well-characterized in-bred strains of mice that have been studied under different nutritional conditions and with different genetic backgrounds. The HMDP is a valuable resource to investigate metabolic and cardiovascular traits and validate systems genetics [57,58,59]. 

For this study, mouse strains on a high-fat diet (hereafter HF mice) were compared to their equivalent strains after transgenic implementation of human APOE-Leiden (apolipoprotein-E-Leiden) and CETP (cholesterol ester transfer protein) (hereafter TG mice), leading to distinct hyperlipidemia and inflammation. Baseline Ldl and Vldl cholesterol levels were significantly higher in TG mice compared to HF mice (42 mg/dL vs. 92 mg/dL; *p* < 0.001). Furthermore, inflammation-associated factors such as Il1b, Il6 and Tnfα were significantly increased in TG mice contrasted to HF mice (*p* < 0.001). To assess the role of Atf3 in HF and TG mice Atf3 expression was correlated to the expression of Ldlr and Ldl/Vldl cholesterol plasma levels. HF mice showed no significant correlation between Atf3 and Ldlr expression (Pearson’s R 0.08, *p* > 0.05), although Atf3 showed a weak positive correlation with Ldl/Vldl cholesterol values (Pearson’s R 0.28, *p* < 0.01). In contrast, in TG mice, which are more prone to hyperlipidemia and inflammation, a significantly inverse correlation between Atf3 and Ldlr expression was apparent (Pearson’s R −0.70, *p* < 0.001). Atf3 expression and Ldl/Vldl cholesterol values were significantly positively correlated (Pearson’s R 0.55, *p* < 0.001).

## 3. Discussion

Functional regulatory networks identified ATF3 as a central regulator of an atherosclerosis relevant liver network based on candidate genes from human CAD GWAS and atherosclerosis relevant genes validated in genetically engineered mouse models. Prediction by Bayesian modeling in combination with in vitro and in vivo validation revealed the hierarchy and potential regulations in this network, highlighting a key role of ATF3 as a novel regulator of LDLR and cholesterol metabolism homeostasis.

Transcriptomic data from humans and mice predicted ATF3 to regulate LDLR directly and via MAFF indirectly based on implemented consensus directionalities. ChIP-seq data of human HepG2 cells supported this assumption as ATF3 binding sites were identified in promoter and intron regions of LDLR and MAFF. 

In experimental model systems and human data, MAFF has already been shown to regulate the expression of LDLR in a context-specific manner. MAFF induces LDLR expression under non-inflammatory conditions. After stimulation by LPS, MAFF downregulated LDLR expression. The underlying mechanism is based on the binding of MAFF heterodimers to the Maf recognition element, also known as a stress-responsive element, in the promoter of LDLR. The preferred binding partner under inflammatory conditions was the CAD-GWAS candidate gene BACH1 [22]. 

Experiments in human Hep3B liver cells support the regulatory impact of ATF3. siRNA knockdown of ATF3 resulted in significant upregulation of MAFF and LDLR. Stimulation of Hep3B cells with LPS caused significant upregulation of ATF3 and MAFF, whereas LDLR was downregulated. The network has been previously shown to be susceptible to inflammation. LPS stimulation of mouse AML12 liver cells resulted in the same effect on MAFF and LDLR expression [22]. In line with the LPS-induced ATF3 upregulation, ATF3 expression has been subject to metabolic, inflammatory and atherosclerosis research before [23,24,25,26,27,60,61,62,63,64,65,66,67]. Specifically, ATF3 was shown to protect against LPS induced inflammation in mice via inhibiting HMGB1 expression. Further, ATF3 knockout in this mouse model subsequently subjected to LPS was associated with elevated expression of circulating IL6, TNFα, and MCP1, and increased risk of death [62].

STARNET, comprising more than 600 CAD patients who have undergone bypass surgery, is currently the most comprehensive study worldwide to integratively investigate genomic and functional aspects at the transcriptome level in CAD patients [56]. Accordingly, STARNET is a tremendously valuable validation tool for cardiovascular studies. Human in vivo data from STARNET include transcriptome data from liver tissues and show a significant positive correlation between the expression of the transcription factor ATF3 and LDLR as well as ATF3 and MAFF. In addition, ATF3 was identified as a central regulator of ‘liver module 15’, a liver supernetwork associated with cholesterol metabolism and inflammation. This supernetwork showed strong correlation with CAD relevant pathways, especially being related to total cholesterol, LDL cholesterol and atherosclerotic lesions. Importantly, most of the individuals undergoing CABG surgery in STARNET were treated with statins, known to lower LDL cholesterol and mediate anti-inflammatory effects. From this point of view, STARNET can be considered a ‘low-inflammatory’ cohort. Interestingly, also HDL cholesterol was correlated with this ATF3 regulated supernetwork. This finding is in line with recent publications reporting that ATF3 in hepatocytes protects against atherosclerosis by regulating HDL cholesterol and bile acid metabolism. Overexpression of human ATF3 in hepatocytes reduced the development of atherosclerosis in Ldlr or ApoE knockout mice on a high-fat diet, whereas hepatocyte-specific ablation of Atf3 caused the opposite effect [25].

The HMDP represents an exciting repository to further investigate metabolic and cardiovascular traits in mouse models within a controlled environment [61,62,68]. Data from HMDP confirmed the relevant role of ATF3 in cholesterol metabolism. In mice with transgenic implementation of human APOE-Leiden and CETP, a phenotype more prone to inflammation, a significant inverse correlation between Atf3 and Ldlr expression was identified in combination with a positive correlation between Atf3 expression and Ldl/Vldl cholesterol values. 

This study incorporates several limitations. The present study represents a snapshot based on currently available information and bioinformatics resources. As the number of genes relevant to atherosclerosis/CAD will increase in the coming years due to ever larger GWAS analyses as well as increasing numbers of genetically modified mouse models, and the number of available transcriptome studies in humans and mice will multiply the patterns of our predicted model might change. Nevertheless, the modeled network predictions are relatively insensitive to changes in the number of candidate genes included, which supports the robustness of the overall approach.

Comprehensive in vivo data from STARNET support the idea that high ATF3 levels are atheroprotective. However, extensive perturbation via knockdown of ATF3 in Hep3b cells led to immediate upregulation of LDLR in vitro. Interpretation of these results should take into account the following considerations. In general, ATF3 is a transcription factor with non-specific binding capacities to a range of interaction partners. As ATF3 binds the cAMP response element (CRE), a sequence present in many cellular promoters it has the capacity to up- or downregulate transcription from promoters with ATF sites. The regulatory impact might be influenced by different effect sizes and functions of interacting dimerization partners, differential splicing, presence of isoforms, environmental perturbation or intake of medication. Specifically, in vitro models were limited to one human liver cell line, defined siRNA incubation times and a relatively high dose of LPS. Further, the molecular mechanisms affecting ATF3 expression and ATF3 downstream targets remain inadequately understood. In particular, molecular heterodimerisation partners of ATF3 influencing transcriptional activation or repression of MAFF and LDLR under different conditions remain unknown. In this regard, this study might be a good starting point and will stimulate further investigations. 

In summary, this study provides novel insights identifying ATF3 as a central regulator of an atherosclerosis relevant liver network that is closely related to inflammation and cholesterol metabolism. Most importantly, ATF3 was identified as direct and indirect (via MAFF) regulator of the LDLR and plays a crucial role in the homeostasis of LDL cholesterol metabolism.

## 4. Materials and Methods

The established bioinformatics approach has already been published before [22]. Very condensed human CAD GWAS and mouse atherosclerosis candidate genes were retrieved from the literature [16,34,35,36,38,39,40,41,42,69,70,71]. Bayesian gene regulatory networks derived from previous expression analyses on human and mouse tissues supported the prediction of gene interactions and their regulatory capacities [68,72,73,74,75,76]. Central regulators of the atherosclerosis relevant liver network were identified based on an established key driver analysis [15,68,77,78,79].

Cell experiments were performed using human Hep3B cells (ATC-HB8064). Cells were cultured in Dulbecco’s modified Eagles medium supplemented with 10% FBS and 1% each penicillin/streptavidin. All experiments were performed in 6-well plates (1.5 × 10^5^ cells/well). For small interfering RNA (siRNA) transfection, cells were treated for 48 h with 100 nmol of ATF3-specific siRNA (Silencer Select, Ambion, Austin, TX, USA) and nontargeting control siRNA (scrbl) (Silencer Select, Ambion, Austin, TX, USA) using Lipofectamine RNAiMax (Lipofectamine RNAiMAX, Invitrogen, Waltham, MA, USA) and OptiMEM medium containing 5% fetal bovine serum (FBS) according to the manufacturer’s instructions. Twenty-four hours after siRNA transfection, 100 ng/mL LPS (Lipopolysaccharide, Sigma, St. Louis, MO, USA) was added to the transfection medium as an inflammatory stimulus. PBS treatment was used for the untreated control group.

Total RNA was extracted from Hep3B cells using a modified protocol of the semi-automated Maxwell^®^ RSC simply RNA tissue kit (AS1340 Promega©). The in-house established protocol included mechanical (Tissue Grinder, NIPPON Genetics EUROPE, Düren, Germany) and chemical as well as enzymatic disruption of the cells. After mechanical homogenization of Hep3B cells in 1-thioglycerol-containing homogenization buffer, cells were lysed and digested in proteinase K-containing lysis buffer, followed by semi-automated RNA extraction according to the company’s protocol. Elution took place in 60 µL of nuclease-free water, and the concentration was measured spectrometrically (Tecan, Männedorf, Switzerland, Lifesciences). The isolated RNA was stored at −80 °C. Reverse transcription-polymerase chain reaction was performed using the High-Capacity RNA-to-cDNA™ Kit (Applied Biosystems, Waltham, MA, USA) with random oligo-dT primers and RNA at a final concentration of <100 ng/µL according to the manufacturer’s protocol (37 °C 1 h; 95 °C 5 min; 4 °C ∞).

Quantitative real-time PCR was performed using TaqMan^TM^ Universal Master Mix II (Applied Biosystems) and TaqMan^TM^ Gene Expression Assay (ThermoScientific, Waltham, MA, USA) for ATF3, LDLR, MAFF and ACTB according to the manufacturer’s protocol (40 amplification cycles). Target expression was standardized by ACTB as a housekeeping gene and normalization took place to the untreated control.

Cell experiments were performed at least four times in duplicates. Results are expressed as mean +/− SD. Statistical analysis was performed with an unpaired t test for comparison of two groups. Bonferroni correction was applied for multiple comparisons. A *p* value of <0.05 was considered statistically significant. Statistical analysis was performed using GraphPad Prism version 9.4.0 (GraphPad Software, San Diego, CA, USA). Further analyses were performed using R version 4.0.55 (R Foundation for Statistical Computing, Vienna, Austria). 

For the identification of ATF3 binding sites free accessible chromatin immunoprecipitation data coupled with sequencing (ChIP-seq) of ATF3 on human liver cell line HepG2 was retrieved from GEO database (accession: GSE169788). Two replicates were processed using ENCODE Transcription Factor ChIP-seq processing pipeline (www.encodeproject.org/chip-seq/transcription_factor (accessed on 3 May 2022)). Significant ChIP-seq signals (*p* value < 10^−5^) were displayed and inspected in Integrative Genomics Viewer (IGV; https://software.broadinstitute.org/software/igv/ (accessed on 3 May 2022)). Peaks were considered as conserved if replicates reached the IDR (Irreproducible Discovery Rate) threshold of 0.05. The MEME web service (https://meme-suite.org/meme (accessed on 3 May 2022)) was used in this study to detect binding site motif patterns of ATF3 in promoter and intron regions of LDLR and MAFF.

In vivo validation was based on human transcription data from STARNET (Stockholm-Tartu Atherosclerosis Reverse Network Engineering Task) [56] and mouse transcription data from the Hybrid Mouse Diversity Panel (HMDP) [61,62,68]. STARNET includes genetic and transcriptomic data from 600 CAD patients (cases) treated by coronary artery bypass grafting (CABG) and 250 patients with angiographic exclusion of CAD (controls). The controls generally required open heart surgery due to high-grade valvular disease [56]. The Hybrid Mouse Diversity Panel (HMDP) comprises 105 well-characterized inbred strains of mice that can be used to analyze genetic and environmental factors underlying complex traits. The HMDP has been—amongst others—studied for traits relevant to atherosclerosis and atherosclerosis relevant risk factors such as obesity, diabetes and inflammation. Therefore, the HMDP is a valuable resource to validate systems genetics analyses of metabolic and cardiovascular traits [61,62,68]. Validation analyses were performed using R version 4.0.55 (R Foundation for Statistical Computing, Vienna, Austria).

## 5. Conclusions

This study demonstrates that the transcription factor ATF3 is a significant key regulator of a dense liver network interacting with a large number of genes known to affect atherosclerosis in humans and mice. ATF3 was predicted to be a direct and indirect regulator of the LDLR. In vitro siRNA knockdown of ATF3 unraveled the regulatory capacity of ATF3 resulting in significant perturbation of the predicted downstream targets LDLR and MAFF. Liver-specific transcription data of human (STARNET) and mouse studies (HMDP) support the relevant role of ATF3 in regulating LDLR homeostasis. We speculate that ATF3 and MAFF, as direct regulators of LDLR, might be suitable candidates for therapeutical approaches to lower LDL cholesterol and reduce long-term cardiovascular risk. 

## Figures and Tables

**Figure 1 metabolites-12-00840-f001:**
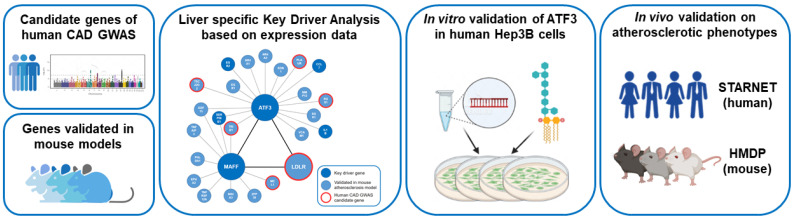
Workflow of the study. Human CAD GWAS candidate genes and atherosclerosis relevant genes validated in genetically engineered mouse models in combination with publicly available transcription data of both species have been used to model liver-specific gene regulatory networks followed by Bayesian Key Driver Analysis (KDA). The focus was on the transcription factor ATF3 orchestrating an atherosclerosis relevant subnetwork including MAFF and LDLR. Validation of the regulatory capacities of ATF3 have been performed experimentally in vitro in human Hep3B cells using small interfering RNA (siRNA) knockdown and in vivo in large transcriptomic studies (STARNET—human and HMDP—mouse). CAD: Coronary artery disease; GWAS: Genome wide association study; HMDP: The Hybrid Mouse Diversity Panel; LPS: lipopolysaccharide; STARNET: Stockholm-Tartu Atherosclerosis Reverse Network Engineering Task.

**Figure 2 metabolites-12-00840-f002:**
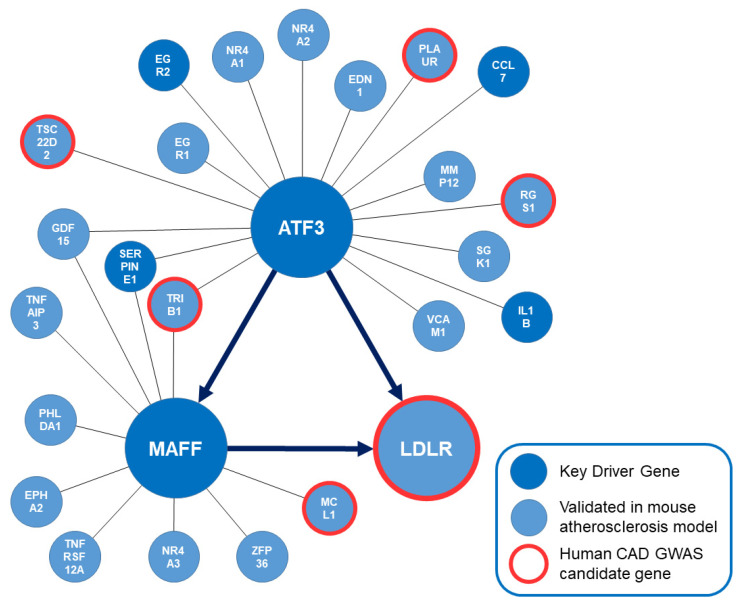
Subnetwork orchestrated by ATF3. Key driver genes of the overall modeling approach are shown in dark blue. Genes that have been validated to affect atherosclerosis in genetically engineered mouse models are shown in light blue. Human CAD GWAS candidate genes are highlighted in red. The bioinformatically predicted directionality of regulation between ATF3, MAFF and LDLR in this liver-specific regulatory network is implemented. ATF3: Activating transcription factor 3F; CAD: coronary artery disease; GWAS: Genome wide association study; LDLR: low-density lipoprotein receptor; MAFF: MAF basic leucine zipper transcription factor F.

**Figure 3 metabolites-12-00840-f003:**
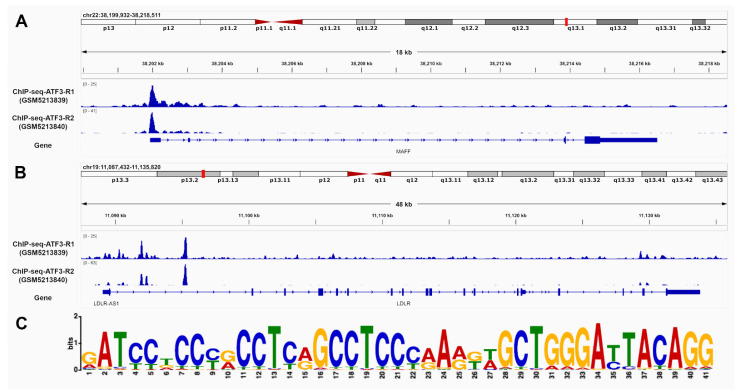
ATF3 has the ability to bind to MAFF and LDLR. Chromatin immunoprecipitation DNA-sequencing (ChIP-seq) data of human HepG2 cells supports potential binding of ATF3 in the promoter and intron regions of (**A**) MAFF and (**B**) LDLR. The matching motif (**C**)was identified using publicly available ChIP-Seq data of the ATF3 gene in human HepG2 cells from ENCODE (Encyclopedia of DNA Elements). The height of the letter represents the frequency of the observed nucleotide in that position. ATF3: Activating transcription factor 3F; ChIP-Seq: Chromatine-immunoprecipitation sequencing; LDLR: low-density lipoprotein receptor; MAFF: MAF basic leucine zipper transcription factor F.

**Figure 4 metabolites-12-00840-f004:**
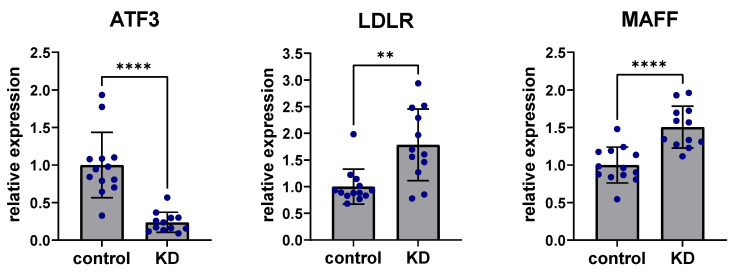
Knockdown of ATF3 perturbs expression of LDLR and MAFF. siRNA knockdown (KD) of ATF3 led to significant downregulation of ATF3 expression levels in human Hep3B liver cells after 48 h. siRNA knockdown of ATF3 significantly upregulated the predicted downstream targets LDLR and MAFF. ** *p* < 0.01, **** *p* < 0.0001. ATF3: Activating transcription factor 3F; KD: Knockdown of ATF3; LDLR: low-density lipoprotein receptor; MAFF: MAF basic leucine zipper transcription factor F.

**Figure 5 metabolites-12-00840-f005:**
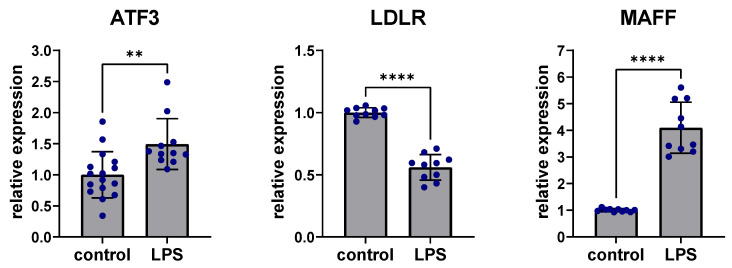
Perturbation of ATF3, LDLR and MAFF expression after LPS stimulation. Stimulation with 100 ng/mL LPS for 24 h led to significant upregulation of ATF3, downregulation of LDLR and upregulation of MAFF in human Hep3B liver cells. ** *p* < 0.01, **** *p* < 0.0001. ATF3: Activating transcription factor 3F; LDLR: low-density lipoprotein receptor; LPS: Lipopolysaccharide; MAFF: MAF basic leucine zipper transcription factor F.

**Figure 6 metabolites-12-00840-f006:**
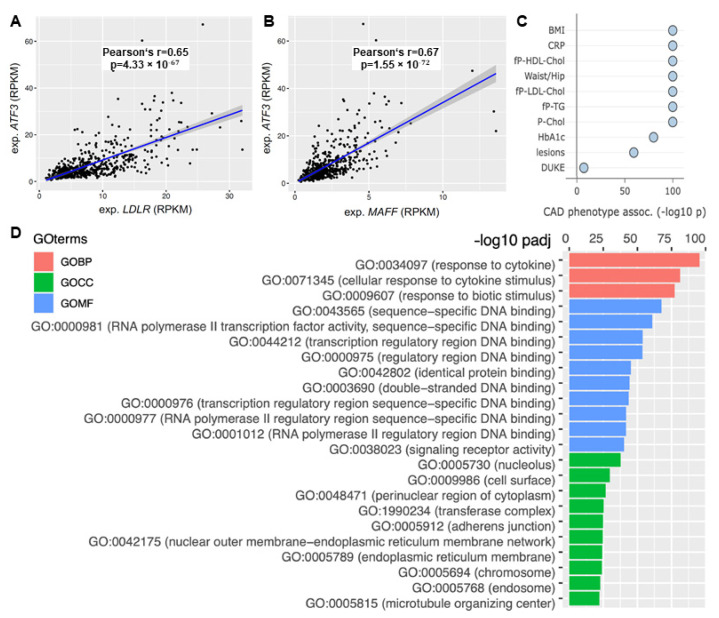
The role of ATF3 as a central regulator of cholesterol metabolism in the human liver. Correlation of ATF3 expression with (**A**) LDLR and (**B**) MAFF expression in STARNET. (**C**) Correlation with CAD associated phenotypes in the liver supernetwork regulated by ATF3 based on expression. (**D**) Pathway analysis based on Gene Ontology terms derived from the liver supernetwork based on expression data. Red bars represent biological pathways, green cellular compartments and blue molecular functions. CAD: Coronary artery disease; GO: Gene Ontology; GOBP: Gene Ontology Biological Pathway; GOCC: Gene Ontology Cell Compartment; GOMF: Gene Ontology Molecular Function; LDLR: low-density lipoprotein receptor; MAFF: MAF basic leucine zipper transcription factor F.

## Data Availability

Data provided in this study are available in persistent repositories. Human data from STARNET (Stockholm-Tartu Atherosclerosis Reverse Network Engineering Task) are accessible through the Database of Genotypes and Phenotypes and the STARNET browser (http://starnet.mssm.edu (accessed on 17 May 2022)). Mouse data from the Hybrid Mouse Diversity Panel (HMDP) are accessible through the Mouse Phenome Database. All experimental data supporting the findings of this study can be requested from qualified researchers at the German Heart Center Munich.

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
