# Peer review of "Identification of the Transcription Factor ATF3 as a Direct and Indirect Regulator of the LDLR"

_metabolites, 2022, doi:10.3390/metabo12090840_

Round 1

Reviewer 1 Report

It’s not clear to me what is declared by the authors in the section entitled “Author Contributions”. Indeed, it is stated that all authors have had a major role in the investigation, which means that all the listed authors including the so-called “DigiMed Bayern Consortium” have participated to the experimental work. This must be explained very carefully. As such, it doesn’t seem realistic that about 100 authors have participated to the experimental work, and the Editorial Office must be aware of a potential issue of inappropriate authorship. Please explain in full details the role on this article of each expert included in the DigiMed Bayern Consortium.

Reviewer 2 Report

The author provided significant evidence supporting that the transcription factor ATF3 has an impact on LDLR expression directly and indirectly. The findings in this manuscript depict a possible molecular mechanism that explains the role of ATF3 in the pathogenesis of cardiovascular diseases. However, a few discrepancies in the results remain to be further discussed. In the abstract (line 45 and 46), the author stated that the knockdown of ATF3 in Hep3B significantly reduced LDLR expression. However, the results presented in Figure 4 show an opposite trend. Moreover, the correlation of ATF3 expression with LDLR and MAFF expression in Figure 6 suggests a positive correlation between them in the human liver. The author needs to explain the discrepancies between in vitro experiments and human data.

Reviewer 3 Report

The paper Metabolites-1868994 » Identification of the transcription factor ATF3 as a direct and indirect regulator of the LDLR« is the result of an extensive international collaboration with contributions from researchers involved in STARNET (Stockholm-Tartu Atherosclerosis Reverse Network Engineering Task), researchers working at the UCLA , the German Heart Centre Munich and several other prominent institutions. This work in an extension of the research on MAFF (MAF basic leucine zipper transcription factor F) as a regulator of atherosclerosis that has been published in Circulation 2021 (DOI: 10.1161/CIRCULATIONAHA.120.050186 ). Candidate genes associated with coronary artery disease were validated in mouse models. Gene regulatory networks, focusing on ATF3 and LDLR were studied by computer modeling /bioinformatics (Bayesian key driver analysis) and validated in cultures of human Hep3B cells, and finally validated in atherosclerotic phenotypes in humans and in mice.

The transcription factor ATF3 was found to be a regulator of a liver network linked to inflammation and cholesterol metabolism, directly and indirectly (via MAFF) regulating LDLR. Upregulation of ATF3 resulted in downregulation of LDLR. This work adds to the knowledge on links between inflammation and atherosclerosis and opens possibilities for testing ATF3 as a therapeutic target.

The paper covers a wide range of experiments and is necessarily at times only schematic, but is well written and balanced. The conclusions are supported by the results. I recommend accepting the paper for publication in Metabolites.

Round 2

Reviewer 1 Report

Thanks for the clarification, and you are now stating exactly what I was saying.

Although I am disappointed that you didn’t revise the authorship according to my suggestion, I still wish to give you another change to make appropriate revision since I appreciate the quality of your work.

The reality is that your manuscript was prepared only by few authors, shared with all members of the consortium, and then reviewed and edited only by some of them. This is what happens in any large international expert panel. I have personally very large experience on that. Still, you reasoned to publish the final version of the manuscript as a consortium, but this is incorrect in principle, since it is likely that the vast majority of these 100 authors didn’t even read the article.

Please revise accordingly. You cannot keep 100 people included in the so-called “DigiMed Bayern Consortium” as co-authors of the present manuscript. You need to include as co-authors only those who actively participated in the experimental work and/or significantly contributed in the writing and the editing. Authorship is a serious thing.

Author Response

We wish to thank Reviewer 1 for the latest review. 

Following the recommendation of Reviewer 1, the DigiMed Bayern Consortium was removed from the list of co-authors, as not all individuals included in the consortium provided experimental input or have taken the provided opportunity to review or edit the manuscript.

Scientific input of the DigiMed Bayern Consortium as a group has now been moved to the acknowledgment section only.

Round 3

Reviewer 1 Report

accept